theoretical biology, synthetic biology, ecology

gene drive, polyandry, invasive species, pest control, house mouse

**Author for correspondence:**
Andri Manser
e-mail: andri.manser@liverpool.ac.uk

One contribution to the Special Feature 'Natural and synthetic gene drive systems'. Guest edited by Nina Wedell, Anna Lindholm and Tom Price

# Controlling invasive rodents via synthetic gene drive and the role of polyandry

Andri Manser[1], Stephen J. Cornell[1], Andreas Sutter[2], Dimitri V. Blondel[3], Megan Serr[3], John Godwin[3] and Tom A. R. Price[1]

[1]Institute of Integrative Biology, University of Liverpool, Biosciences Building, Liverpool, UK
[2]Centre for Ecology, Evolution and Conservation, University of East Anglia, Norwich, UK
[3]Department of Biological Sciences, North Carolina State University, Raleigh, NC 27695-7617, USA

AM, 0000-0002-4296-7434; AS, 0000-0002-7764-3456; MS, 0000-0001-7599-6605

House mice are a major ecosystem pest, particularly threatening island ecosystems as a non-native invasive species. Rapid advances in synthetic biology offer new avenues to control pest species for biodiversity conservation. Recently, a synthetic sperm-killing gene drive construct called *t-Sry* has been proposed as a means to eradicate target mouse populations owing to a lack of females. A factor that has received little attention in the discussion surrounding such drive applications is polyandry. Previous research has demonstrated that sperm-killing drivers are extremely damaging to a male's sperm competitive ability. Here, we examine the importance of this effect on the *t-Sry* system using a theoretical model. We find that polyandry substantially hampers the spread of *t-Sry* such that release efforts have to be increased three- to sixfold for successful eradication. We discuss the implications of our finding for potential pest control programmes, the risk of drive spread beyond the target population, and the emergence of drive resistance. Our work highlights that a solid understanding of the forces that determine drive dynamics in a natural setting is key for successful drive application, and that exploring the natural diversity of gene drives may inform effective gene drive design.

## 1. Introduction

Rodents such as mice (*Mus musculus*) and rats (*Rattus rattus* and *Rattus norwegicus*) are major pest species. As a result of human activity, they have successfully invaded nearly every landmass across the globe. They are particularly detrimental to island ecosystems where they cause great damage to endemic fauna as non-native invasives [1–4]. Current methods to control invasive rodent populations via trapping, poisoning or habitat management have produced considerable conservation gains [5], but also have significant limitations. The use of anticoagluant toxicants, currently the main method to control populations, often results in the killing of non-target species, and the slow internal bleeding induced by the toxin raises major animal welfare concerns. Moreover, toxicants pose a risk to human health on inhabited islands, which constitute most of the islands where rodents currently pose an ecosystem threat [6]. Overall, the negative impacts of invasive rodents (loss in biodiversity) currently seem to out-pace our ability to control populations [7].

Recently, there has been great excitement around the possibility of using synthetic gene drives as a tool for pest control in general [8,9], and for biodiversity conservation in particular [3,10]. Gene drives are genetic elements that manipulate reproductive processes to gain a transmission advantage over the rest of the genome. This often occurs through the distortion of meiosis or gamete development (termed 'meiotic drive'), or by breakage and self-insertion into the homologous target sequence (termed 'homing-based drive', [11]). As a result, gene drive systems are transmitted to subsequent generations at a

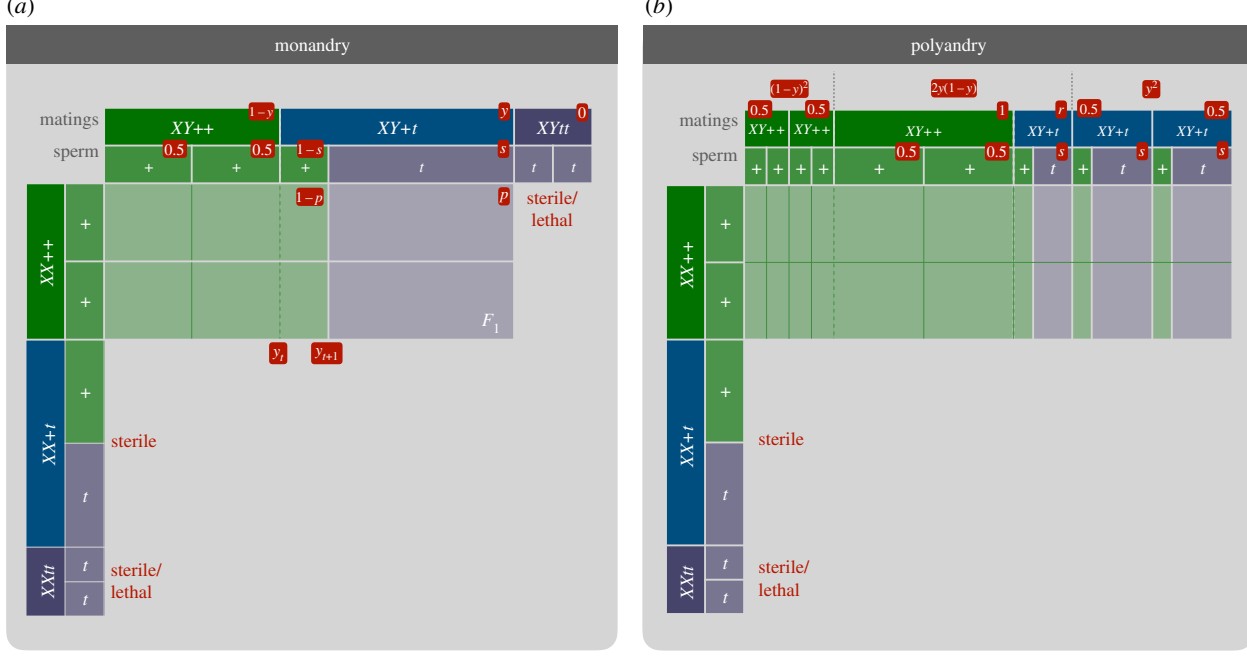

**Figure 1.** Schematic illustration of mating/reproduction in the *t-Sry* system under (*a*) monandry and (*b*) polyandry. Both figures represent classical mating tables with 'females'/the three *XX* genotypes and their ova production in the first two columns, respectively, and their male mating partners and their sperm production in the first two rows, respectively, such that their projected intersection corresponds to the $F_1$ offspring genotypes produced. Green and violet colours denote wild-type and drive organisms/gametes, respectively. Probabilities of respective matings and fertilizations are provided in the red boxes (parameters: drive genotype frequency $y$, drive strength $s$, and drive male sperm competitiveness $r$). Under (*a*) monandry, the drive frequency among the $F_1$ offspring ($y_{t+1}$), which is equivalent to the probability that a female is fertilized by a drive sperm $p$, is at best identical (if drive is complete) but typically smaller than the drive genotype frequency among parents $y_t$. In (*b*) where all females mate with two males (polyandry), probabilities of the three male combinations are additionally provided in red boxes at the top. The probability of drive fertilization $p$ (equivalent to violet surface) is reduced further if drive males are inferior at sperm competition ($r > 1$), but depends on the frequency of cases where wild-type and drive males compete ($2y(1-y)$). (Online version in colour.)

frequency greater than the 50% expected by Mendelian inheritance. The super-Mendelian mode of transmission allows drive systems to rapidly spread though populations, even in scenarios where they confer substantial fitness costs to the carrier organism as a whole [12]. As a result, gene drives have been proposed as a means to spread genes with useful properties for human ends into natural populations (population replacement) or to eradicate a pest population entirely (population suppression, see this paper for an example). Rapid advances in synthetic biology make the large-scale use of synthetic drive at an affordable price a real possibility, and could revolutionize the way humanity deals with pests and diseases. However, gene drive technology poses fundamental and new challenges, ranging from understanding the ecological and evolutionary impacts of a release [13], risks and benefits, to considerations about the ethics and regulation surrounding the technology [14,15].

One possibility to control pest species via artificial gene drive is the release of sex-linked drivers that will eradicate a target population owing to a lack of one sex. Recently, such a drive construct called *t-Sry* has been proposed as a means to control invasive populations of house mice (*M. musculus*, [3,10]). The *t-Sry* is a combination of two genes (complexes): (i) the *t* haplotype, a naturally occurring autosomal meiotic driver, and (ii) the male-determining *Sry* gene, which is normally found on the Y chromosome. The *t* haplotype is a sperm killer that damages sperm development of non-*t* bearing wild-type sperm (+) in heterozygote males. As a result, heterozygote +/*t-Sry* males transmit the construct to 90–99% of their progeny instead of the 50% expected under

Mendelian inheritance. The *Sry* gene will then cause XX *t-Sry* genotypes, which would normally develop as females, to phenotypically develop as males. These individuals will also be *sterile* because they lack essential Y-linked genes for sperm development. In summary, drive males will pass on the *t-Sry* to nearly all offspring, which will result in more drive males (half fertile and half sterile), but, importantly, none or very few reproductive females (see figure 1 for an schematic overview, also [10]). Efforts are currently underway to determine whether release of *t-Sry* males could eliminate populations of invasive mice on islands through a lack of reproductive females, thereby protecting local biota. Recent modelling has suggested that repeated releases of *t-Sry* males could indeed be used to overwhelm a population, radically reducing the number of females, causing population collapse and eradication [16].

Despite the great potential, recent theoretical and empirical studies have highlighted that the rapid evolution of resistance to gene drive could pose a serious problem for drive technology. For example, individuals which carry genes that interfere with the drive mechanism at a molecular level (suppressor genes) will no longer suffer from some of the costs associated with the driver. As a result, resistance is expected to rapidly sweep through a population [17–19], a prediction that has been corroborated in both laboratory and natural populations [20,21]. The problem of resistance evolution appears particularly salient in homing-based gene drive systems which typically attack a rather specific target sequence (such as CRISPR/Cas9 which is currently most widely discussed as a genetic tool to control pests [18]).

Synthetic constructs that make use of naturally occurring drive systems, such as the *t-Sry* discussed here, may have the critical advantage that the drive mechanism has already 'survived' coevolution with the host genome for considerable time periods. The *t* haplotype has been present in mouse populations around the world for over 2 Myr [22], and, surprisingly, there is little to no evidence of suppressor genes that prevent the sperm killing [23].

Yet even in drive systems where the likelihood of direct molecular suppression is low, drive spread may be hampered by other mechanisms. In the context of the *t-Sry* system, several such indirect non-molecular drive suppression mechanisms that could hamper the success of a release campaign are currently being discussed [3,24]. There is the possibility for survival differences between native and introduced mice, either because the *t-Sry* construct itself affects survival, or owing to differences in the genetic background (if *t-Sry* is bred up for release on a different genetic background). It is currently unknown how both factors will affect the fitness of *t-Sry* carriers. The *t* haplotype comprises about 1.5% or 40 Mb of the mouse genome that is protected from recombination through inversions. As a result, one would expect the accumulation of deleterious mutations. However, apart from strong *t/t* homozygote costs (which are of little relevance for the *t-Sry* construct, see below), measures of survival differences between $+/t$ heterozygotes and $+/+$ wild-types in natural populations are inconclusive, with some studies reporting a fitness cost to *t* carriers [25], while other studies find a $+/t$ heterozygote advantage [26,27]. Similarly, if the genetic background comes from a highly competitive mainland strain, it might give the *t-Sry* construct a substantial advantage. Moreover, the genetic background and *t-Sry* might also affect behavioural outcomes, such as social dominance, mating success or dispersal [28–31]. Hence, the impact of *t-Sry* on survival and mating success is currently unclear and potentially complex.

A factor that has so far been underappreciated, but is particularly likely to reduce the spread of a sperm-killing gene drive such as the *t-Sry*, is females mating with multiple males (polyandry) allowing sperm competition to occur. The *t* haplotype gains its transmission advantage by impairing the sperm function of + sperm, i.e. half the drive male's sperm [32]. This causes the *t* haplotype to be transmitted to more than 90% of the offspring in a monandrous mating [30]. However, the damage to the sperm of $+/t$ males makes these males extremely poor when their ejaculate competes with that of a wild-type male. Two laboratory studies on two independent mouse populations have found that *t* haplotype carrying males only fertilize about 20% of offspring when competing against wild-type males [33,34]. Moreover, genetic data from natural populations suggest house mice are polyandrous, with genetic measures suggesting that 20–30% of wild females produce litters that have multiple fathers [35,36], with similar rates seen in island populations of mice [37]. Hence the substantial disadvantage of *t* haplotype carriers in sperm competition is likely to have important ramifications for the *t-Sry* system. In fact, polyandry is currently our best explanation for the relatively low observed *t* frequencies in natural populations, which typically vary between 5 and 30% [27,38]. Moreover, in the context of the *t-Sry*, the increased male bias in a population is likely to increase the number of matings by females, and might thus intensify the impact of sperm competition.

Indeed, both theoretical work [27,39], and practical work in insects [40] suggest that sperm competition can prevent the spread of sperm-killing drivers, and can determine their frequency in a population [41].

Given the theoretical and experimental evidence highlighting the impact of polyandry on *t* drive, we here evaluate the impact of sperm competition on a *t-Sry* release campaign by means of a theoretical model. In particular, we ask: (i) how the ecological dynamics following a *t-Sry* release are affected by different levels of multiple mating and drive male competitiveness, and (ii) how sperm competition affects the number of animals that have to be released to successfully eradicate a population.

## 2. The model

To investigate the ecological consequences of a *t-Sry* driver release into a mouse population, we consider survival and reproduction based on two loci. The *t-Sry* allele, henceforth abbreviated as *t*, and the homologous wild-type allele + segregate at the drive locus. The conventional sex chromosomes $X$ and $Y$ segregate at a non-linked locus. In principle, we thus have six possible diploid genotypes ($XX++$, $XX+t$, $XXtt$, $XY++$, $XY+t$, $XYtt$). However, matters are simplified considerably by the fact that $XXtt$ are sterile or non-viable (depending on the *t* haplotype variant deployed [42]) and $XX+t$ individuals are sterile, leaving only $XX++$ genotypes as potential mothers (figure 1). Because *all* reproductive females are $XX++$, $tt$ homozygotes cannot feature at any life stage. This leaves us with four genotypes to track- $XX++$, $XX+t$, $XY++$ and $XY+t$. For simplicity, we further assume that the sex chromosomes are selectively neutral, i.e. there are no viability differences between $XY$ and $XX$ individuals (see the electronic supplementary material, text S4 for an analysis where this assumption is relaxed). We can now capture the ecological dynamics of the entire system by considering three state variables: the total population size $N$, the number of wild-type females $W$ ($XX++$) and the number of *t-Sry* carrying males $D$ ($XY+t$, $D$ for drive). Because $XY$ and $XX$ have the same survival rates, the number of wild-type males will also be $W$ ($XY++$). Finally, the number of sterile $D'$ ($XX+t$) individuals is given by $D' = N - D - 2W$.

### (a) Ecological dynamics

Here, we model an intervention on an island population of mice by following the change in density of the three state-variables ($N$, $W$, $D$) in continuous time. In general, we assume the island has a population of mice at carrying capacity and a 1 : 1 sex ratio. We then add male *t-Sry* mice to the population, either once, or repeatedly at rate $\mu$. Both represent reasonable strategies currently under consideration for the use of *t-Sry*. We assume that all genotypes die with a baseline mortality rate $m_1$ (see the electronic supplementary material, text S4 for an examination of differential survival). To introduce an upper limit to population growth (carrying capacity $K$), we further assume an additional death rate $m_2$ that increases with the overall population density $N$. Females $W$ give birth to new mice at birth rate $b$. Finally, the key parameter of the model $p$ denotes the probability that a given offspring is fertilized by a drive male (which depends on sperm competition, see next section). New wild-type $++$ individuals are produced if

reproductive females are fertilized by wild-type sperm (at probability $1 - p$), of which half will be female $(1 - p/2)$. Equivalently, reproductive females produce drive males with probability $p/2$. Based on these assumptions, we have the following system of ordinary differential equations (the basic model is equivalent to previous modelling work on the system by Backus & Gross [16]):

$$
\left.\begin{aligned}
\frac{\mathrm{d}N}{\mathrm{d}t} &= bW - (m_1 + m_2 N)N + \mu, \\
\frac{\mathrm{d}W}{\mathrm{d}t} &= bW\frac{1-p}{2} - (m_1 + m_2 N)W \\
\text{and} \quad \frac{\mathrm{d}D}{\mathrm{d}t} &= bW\frac{p}{2} - (m_1 + m_2 N)D + \mu.
\end{aligned}\right\} \quad (2.1)
$$

### (b) Sperm competition

To examine the effects of polyandry and sperm competition on the probability that an egg is fertilized by a *t-Sry* sperm $p$, we here consider two scenarios, a null model where all females are monandrous and a simple polyandry model where females mate either with one or two males (figure 1).

### (i) Monandry model

Let us first define the frequency of drive males among all (fertile) males, $y = D/D + W$. The probability that female ova will be fertilized by a drive sperm in a scenario of single, random mating ($p_1$, where 1 denotes the number of male mating partners) is simply the probability of a mating with a drive male ($y$) multiplied by the probability that a drive sperm is successful ($s$, see figure 1a for a graphical representation). We have $p_1 = ys$. Parameter $s$ measures the strength of meiotic drive, with $s = 0.5$ corresponding to Mendelian inheritance and $s = 1$ to complete drive. For an examination of a non-random mating scenario where females avoid mating with drive males, see the electronic supplementary material, text S4.

### (ii) Polyandry model

In the polyandry model, we assume that a female either mates once (at probability $1 - \psi$) or twice (with probability $\psi$) during an oestrous cycle (see the electronic supplementary material, text S1 for a description of the case where a female mates with an arbitrary number of males). Parameter $\psi$ hence measures the level of polyandry in the population. Crucially, we assume that drive males have a sperm competitiveness of $r$ relative to wild-type males (whose sperm competitiveness equals unity). As a result, a drive male will only fertilize $r/(1 + r)$ eggs when in sperm competition against a wild-type male. For example, if a drive male's sperm competitiveness is half relative to a wild-type male ($r = 0.5$), he will only fertilize one-third of a female's eggs when competing against a wild-type male. There are now three ways in which a female will see her eggs fertilized by a drive male. First (as above), a female may mate with a single drive male (where the probability of fertilization by drive sperm equals $(1 - \psi)ys$). Second, she can mate with two males of which both are drive carriers (where the probability of drive fertilization equals $\psi y^2 s$). Third, she can mate with both a drive and a wild-type male (in which case the probability of fertilization by drive sperm equals $\psi 2y(1 - y)s\, r/(1 + r)$, see figure 1b for a graphical representation). If we sum over these three

outcomes and simplify, we get

$$
p_2 = ys\left(1 - \psi(1 - y)\frac{1 - r}{1 + r}\right). \quad (2.2)
$$

The subtraction term in the bracket measures the *reduction* in drive fertilization probability that is owing to sperm competition. As one would expect, sperm competition only plays a role if females are polyandrous ($\psi > 0$) and drive males have a sperm competitive disadvantage ($r < 1$). Moreover, note that the impact of sperm competition depends on the frequency of *t-Sry*. It is highest when the frequency of drive males $y$ is low, because the drive males will mainly encounter wild-type males as sperm rivals (to which they are inferior), and lowest if drive frequency is high where drive males predominantly compete against other drive males.

## 3. Results

For the purposes of this paper, we examine a scenario with a population at carrying capacity with males and females at equal frequency. We then examine the ecological and evolutionary consequences after (A) a single and (B) a continued release of *t-Sry* males into the population. We consider the population eradicated if the number of reproductive females drops below one ($W < 1$).

### (a) Single release

We first examine a scenario where a number of animals $D_0$ are released into the population *once*. We thus examine the dynamics of the system with $\mu = 0$ (no continued release) and starting conditions $\{N = N_0 + D_0, W = N_0/2, D = D_0\}$.

It is relatively straightforward to see why *t-Sry* cannot spread in a population if only released once into a population (see figure 1 for a geometrical/graphical illustration of this argument). At any point in time, the frequency of drive-carrying animals is given by $y = D/D + W$. The number of drive carriers among newborns, on the other hand, is only $p_1 = ys$ or $ys(1 - \psi(1 - y)(1 - r)/(1 + r))$ under monandry or polyandry, respectively, which will be smaller than the current levels of drive ($p_2 \leq p_1 \leq y$), at least as long as drive is incomplete ($s < 1$) and drive males are equal or inferior sperm competitors ($r \leq 0$), which will almost always be the case in reality. Under monandry, for example, the best a driver could achieve is to maintain the same frequency as the previous generation, but only if drive is complete ($s = 1$).

More formally, the system equation (2.1) will reach a long-term equilibrium $\{\hat{N}, \hat{W}, \hat{D}\}$ if $\mathrm{d}N/\mathrm{d}t = \mathrm{d}W/\mathrm{d}t = \mathrm{d}D/\mathrm{d}t = 0$. In the monandry model, the system of equations has three solutions, of which only $\hat{N} = (b/2 - m_1)/m_2$, $\hat{W} = (b2 - m_1)/2m_2$, $\hat{D} = 0$ and $\hat{N} = \hat{W} = \hat{D} = 0$ are biologically feasible (see the electronic supplementary material, text S2). In both cases, the driver is absent ($\hat{D} = 0$). Henceforth, we can use the population equilibrium in absence of the driver as our carrying capacity $K := \hat{N} = (b2 - m_1)/m_2$. Electronic supplementary material, figure S5 shows a number of numerical solutions of population dynamics, illustrating that a single release is insufficient to eradicate the population in the long term.

### (b) Continued release

We have seen that population eradication is not possible if drive males are only released into the population once, even in the absence of sperm competition. How do the

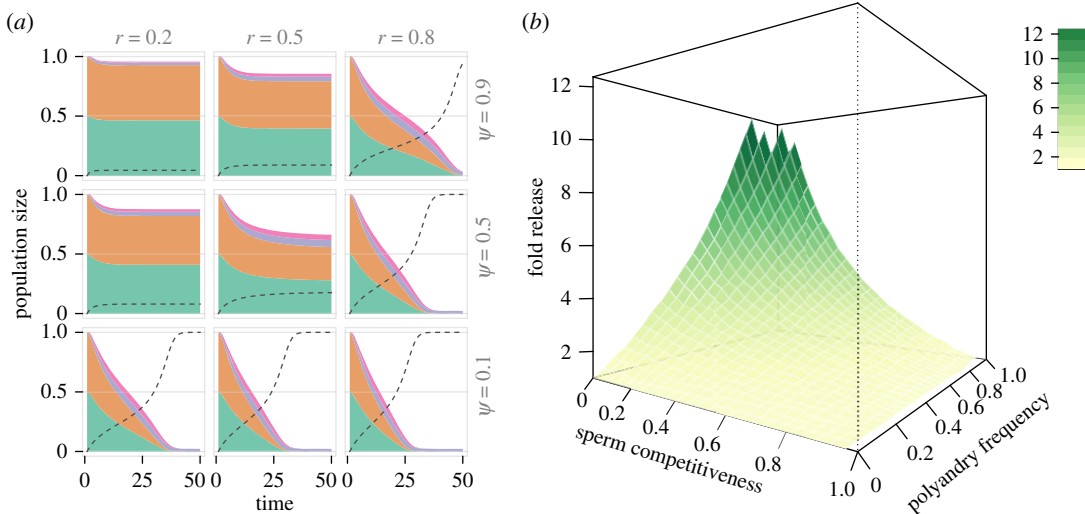

**Figure 2.** (a) The number of wild-type males and females (W in green and orange), t-Sry males (D in violet) and t-Sry 'females' (pink) as a proportion of carrying capacity K when drive males are released continually into the population ($\mu_{std} = 0.02$). The dotted line represents the drive frequency y in the population. The nine panels represent trajectories for different levels of sperm competitiveness r and polyandry levels $\psi$, keeping all other parameter values (including release effort) constant. (b) The minimal release effort $\mu_{std}^{\star}$ compared with baseline (without sperm competition) required to eradicate the population as a function of drive male sperm competitiveness r and polyandry rate $\psi$. Remaining parameter values for both subfigures: $s = 0.9$, $b = 3$, $m_1 = 0.9$, $K = 1000$. (Online version in colour.)

dynamics change under a continued release of drive animals into a population ($\mu > 0$)? To make release effort parameter $\mu$ meaningful from a practical perspective, we express it relative to population carrying capacity K, $\mu_{std} = \mu/K$. For example, if the carrying capacity is $K = 1000$ and $\mu_{std} = 0.01$, 10 drive carriers are released at each given time interval $\Delta t$. We then calculate the minimal release rate $\mu_{std}^{\star}$ required to push the population to eradication ($W < 1$ when $\mu > \mu_{std}^{\star}$). Analytical examination of the full equation system is possible in principle, but results in complicated expressions that offer little insight (but see the electronic supplementary material, text S3 for some approximate results). To derive $\mu_{std}^{\star}$ for a given parameter combination, we numerically calculated release dynamics for 20 different values of $\mu_{std}$ ranging between 0 and 0.2 (sufficient for all parameter values explored), and recorded the release level $\mu_{std}^{\star}$ where the density of reproductive females drops below the eradication threshold ($W < 1$).

### (i) Monandry model

We first consider the monandry model scenario where females only mate with one male. In this case, the probability that a female's ova are fertilized by a drive male is $p_0 = sy$ (equation (2.2)). We explored parameter dependence by calculating $\mu_{std}^{\star}$, the minimal release rate required to eradicate the population, for systematically varying values of all model parameters: birth rate b, death rates $m_1$ and $m_2$ and drive levels of $s = 0.9$ (the empirical value for nearly all t haplotypes). Note that, because release rate $\mu_{std}$ is expressed relative to carrying capacity K, density-dependent death rate $m_2$ does not affect outcomes. Electronic supplementary material, figure S2 shows release thresholds $\mu_{std}^{\star}$ values as a function of baseline birth and mortality rates b, $m_1$ and s. The necessary release effort in the absence of sperm competition varied between 1 and 15% of carrying capacity. As one would expect, larger efforts are required when drive strength s is low, and baseline reproductive rates $b - m_1$ are high. We also calculated relatively simple critical release thresholds $\mu^{\star}$ based on approximation (see the electronic supplementary material, S3 and figure S2), which may aid to calculate

approximate release rates for a given set of demographic parameters and drive strength without the need of simulation.

### (ii) Polyandry model

Now that we have calculated the required release effort in the absence of sperm competition, let us examine how polyandry ($\psi$) and sperm competitiveness (r) affect a release campaign. Figure 2a shows different trajectories over time for varying levels of sperm competitiveness r and polyandry rate $\psi$, holding all other parameters constant. It illustrates that sperm competition has strong effects on the predicted dynamics—the same release rate can be sufficient for population eradication with some levels of r and $\psi$, but insufficient if polyandry rates are increased or sperm competitiveness of drive males is decreased.

Figure 2b shows how much $\mu_{std}^{\star}$ has to be increased based on different levels of sperm competitiveness and polyandry (compared to the monandry scenario, see above). In general, a bigger release effort is required if polyandry rates are higher and drive male sperm competitiveness is lower, with release efforts increasing up to 12-fold in extreme scenarios. For empirically realistic values of sperm competitiveness (r = 0.2) and polyandry ($\psi \approx [0.2, 0.6]$) the release effort has to be increased three- to sixfold.

### (iii) Ecological Impact

We examined the ecological implications of this increased release effort by calculating the maximum population size $N_{max}$, the required time until eradication, as well as the total number of animals that need to be released for different polyandry levels (figure 3, analogous to [16]). We found that both the overall ($N_{max}$ typically around 105%) as well as the additional ecological burden owing to polyandry ($N_{max}$ is $\approx$ 1% higher when polyandry is included) were relatively small. Total release effort and time until eradication, on the other hand, are more sensitive to polyandry levels. Note that there are release levels (releasing 20% of the total population size for the example used in the electronic supplementary material, figure S5) that minimize (a) the

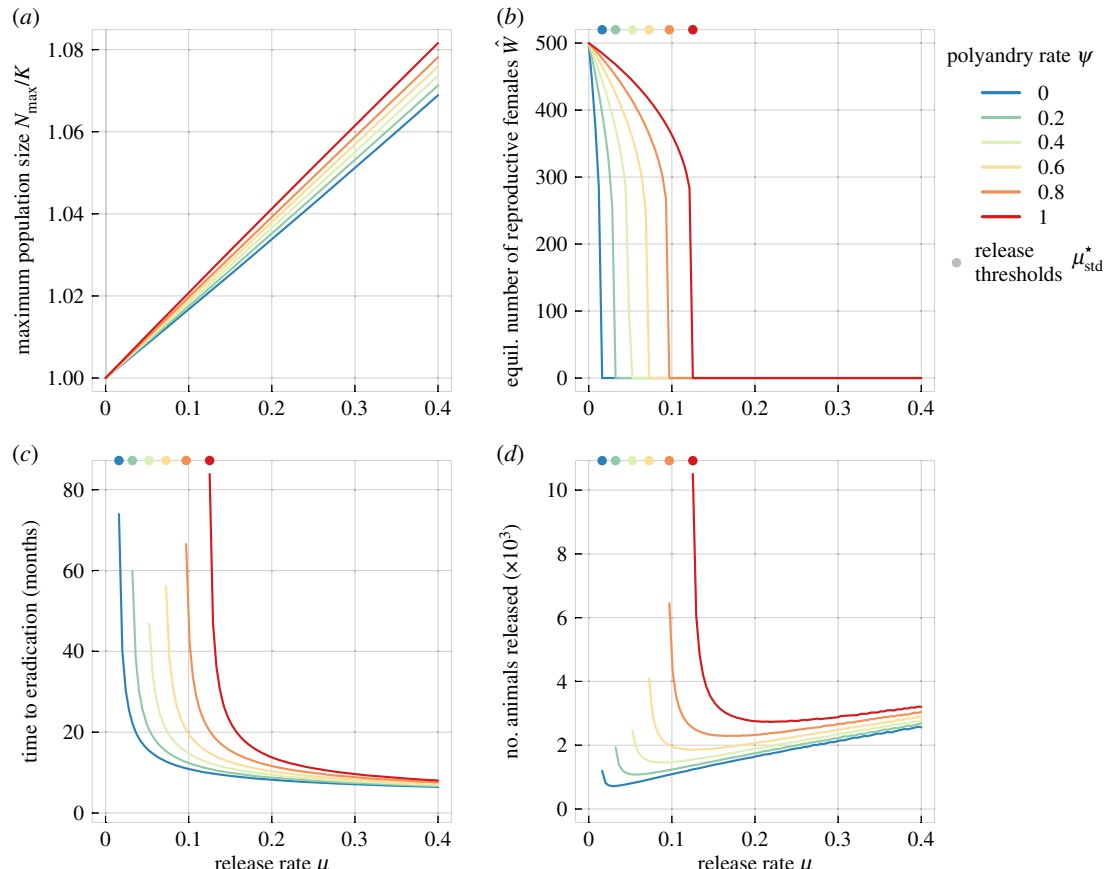

**Figure 3.** The ecological impact of a *t-Sry* release campaign depending on release rates $\mu_{std}$ for different polyandry scenarios (coloured lines, $\psi = [0, 0.2, 0.4, 0.6, 0.8, 1]$) as measured by (a) the maximum population size during a release campaign ($N_{max}$), (b) the equilibrium number of reproductive females after 100 months, (c) the time to eradication, and (d) the absolute number of drive males released. The coloured dots on the top of panels (c,d) denote the minimal release effort required for population eradication $\mu_{std}^{\star}$. The carrying capacity $K$ was assumed to be 1000 and drive male sperm competitiveness was $r = 0.2$ (as typically observed empirically). Remaining parameter values for all panels: $s = 0.9$, $b = 4$, $m_1 = 1$. (Online version in colour.)

total number of animals that need to be released (in line with [16]) and (b) the impact of polyandry, resulting in population eradication in roughly 1 year. However, the precise values will depend on the details of the population demography, i.e. population turnover, and the strength of density dependence.

## 4. Discussion

We investigated the impact of polyandry and sperm competition on a potential eradication campaign using the *t-Sry* gene drive system by means of a theoretical model. First, and in line with a previous theoretical study [16], we find that a single release is typically insufficient to eradicate a population. The driver is hence not self-propagating, which may prevent it from spreading uncontrollably beyond the target population. Second, we show that polyandry and sperm competition pose a considerable quantitative hurdle for a successful release campaign, by increasing the required release effort for a successful eradication three- to sixfold. This strongly suggests that non-molecular factors will impair the success of synthetic gene drive systems, and will need to be considered when planning interventions.

Our model shows that the *t-Sry* construct will not spread through a population if only released once, at least in the absence of fitness differences between *t-Sry* and wild-type mice. This result supports an earlier theoretical

study on the system [16]. However, the inability of *t-Sry* to propagate after a single release can be viewed as a major strength of the system. One of the big risks of gene drive systems is accidental release or uncontrolled spread. An entirely self-propagating drive system, in an organism that has a history of extremely rapid spread and colonization, would be extremely difficult to control, and serious ethical concerns about the development and deployment of such systems have been raised [43]. Mechanisms have been discussed that would confine drive spread by altering a gene drive system so it has uncontrollable initial spread, but then halts or is eliminated (e.g. daisy chain drive [44], underdominant gene drive [45] or targeting locally fixed alleles [46]). However, using a gene drive system that cannot spread without repeated releases is inherently safer, and so may be more useful for local interventions, such as systematically eliminating island populations of pest mice. Moreover, it makes the intervention reversible, because the low fitness of the *t-Sry* system would lead it to naturally be lost from the target population if *t-Sry* releases stop before the population is eliminated.

The inability of *t-Sry* to spread after a single release is predicated on the assumption that released and resident animals do not differ in survival, mating success and other measures of fitness not directly related to drive. Backus & Gross [16] analysed the role of *t-Sry*-related survival differences, and showed that drive animals would require a considerable survival advantage (of about 20% under the

scenario presented) over wild-type animals in order for the *t-Sry* to spread in a self-sustaining fashion. We here also examined various survival costs and mate choice against drive carriers, showing that they will increase required release efforts (electronic supplementary material, text S4). However, as discussed in the Introduction, previous work on the survival and mate choice with respect to *t* haplotypes has been inconclusive at best [25,27,30,47,48]. Moreover, it is currently unknown how an additional copy of the *Sry* gene will impact on fitness. Genetic backgrounds that differ from the target population, such as mice derived from a large mainland population, may also raise or lower the fitness of released mice compared to target mice. One possibility would be to release *t-Sry* males derived from a population that has high male competitive success, to increase the success of the released males in gaining matings. However, this carries the risk that recombination may soon introgress the high fitness background into the population, potentially leaving behind a more viable pest population on the island. It is currently unknown whether recombination would be quick enough to introgress into the target population before eradication, but the question warrants further theoretical exploration.

Our model finds that the success of a *t-Sry* release programme will be strongly affected by both the relative sperm competitiveness of *t-Sry* males and the level of polyandry in the target population. This supports theory suggesting that any drive system which impairs sperm competitiveness (e.g. cytoplasmic incompatibility in *Wolbachia*, sperm-killing drivers, chromosome shredders acting during spermatogenesis) may be disadvantaged in polyandrous target species [49]. In house mice, there have been two solid estimates of the sperm competitiveness of *t* males, from one Australian and one Swiss population, both find that *t* males father as few as 20% of offspring when competing with non-*t* males [33,34]. These studies suggest that the sperm competitiveness of *t-Sry* may not differ much between populations, although this is speculative. The frequency of litters with multiple genetic fathers in natural mouse populations have been estimated at around 20–30% [35–37], but can vary substantially between populations, hence measuring polyandry rates in target populations is advised. Also note that all the measures will probably be an underestimate of the actual (behavioural) polyandry rates, because not every mating results in a successful fertilization. Importantly, a *t-Sry* intervention will cause major disruptions in population density and sex ratio, both of which are likely to impact strongly on polyandry rates and thus on the intensity of sperm competition. While rates of polyandry are likely to go down as population densities decrease [35], thereby increasing the efficacy of *t-Sry*, polyandry is likely to increase as the population becomes biased towards males. The precise impact of these two potentially opposing forces on release dynamics will depend on the exact shape of the relationship between density and sex-ratio on polyandry. This information is currently not available for mice, and so here we assumed that the rate of polyandry did not change with population size and sex ratio. Perhaps the best way to examine this issue would be controlled experiments that measure mating behaviour and the intensity of sperm competition for independently varying degrees of density and sex ratio.

Although polyandry increases the number of *t-Sry* mice that need to be released to cause population elimination,

this is unlikely to be a problem that would make *t-Sry* interventions impossible. Firstly, the ecological impact of the releases is unlikely to be high, as even our highest modelled releases did not increase population size by more than 10%. In terms of effectiveness, it is simply a matter of increasing the release effort to overcome the problem of poor *t-Sry* sperm competitive ability. Of course, the effectiveness of the *t-Sry* system ultimately depends on whether its benefits outweigh the costs in a real eradication attempt. Compared to current poison techniques, the *t-Sry* system has the major advantages of negligible damage to non-target species, being more humane, and no real risk to human health. Moreover, the construct may find broader political and ethical support because it only uses genetic material of the target species, making it non-transgenic. However, it may require continuous small releases over the course of several years. By contrast, current poison bait extermination techniques typically involve very high effort baiting and monitoring, but over a relatively short period of time [50]. It is possible that a *t-Sry*-based approach might be most effective on large islands with human populations, where the costs of poison baiting may be high, and the cost of setting up a *t-Sry* mouse production facility may be relatively low.

The conclusions of our model contrast with the more fundamental issues raised in some other synthetic gene drive systems. Just this year, a functional synthetic homing-based drive based on CRISPR/Cas9 has been presented for female mice [51]. However, modelling and experimental work on homing-based gene drive systems in insects (e.g. CRISPR/Cas9) has found extremely rapid evolution of molecular resistance in target populations, either via the evolution of alterations to target sites, or by the involvement of mechanisms that directly impair the function of the CRISPR/Cas machinery in the host [18–20], thereby rendering the driver completely dysfunctional. Harnessing ancient successful gene drive systems such as the *t* haplotype may represent a promising alternative. Rather than attempting to build novel successful gene drives bottom-up, this approach effectively reverse engineers ancient drive systems that have avoided suppression, and transfers their mechanisms to target species. *t* haplotypes have succeeded in maintaining strong drive in very large populations of all *Mus* subspecies, across varied habitats, in spite of 2 Myr of selection on the host genome to suppress it [22,23]. Adapting a natural driver for use in its host species is certainly possible, if a target species has a suitable driver, although such a driver would still be vulnerable to mutation of the cargo. Alternatively, it may be possible to construct synthetic drive in a target species based on a natural drive system in a related species. This suggests that screening target pest species and their close relatives for natural drivers may reveal useful gene drives that can be modified (see [52] for a recent example in mice). Moreover, even if no suitable natural drivers are identified for a target species, learning more about the mechanisms of ancient natural drive systems that have stood the test of time may be highly productive for designing synthetic drive, as it can help us identify potential drive mechanisms that are less vulnerable to resistance evolution. It is possible that some ancient drive systems may target aspects of gametogenesis that cannot be bypassed easily, making it hard or impossible to evolve suppression. We currently have a extremely fragmented picture of the natural diversity and distribution of drivers, but there are already

striking differences in the degree of resistance, ranging from systems with no apparent suppressors (such as the *t* haplotype or *Sex-Ratio* in *Drosophila subobscura*) to systems with various forms of suppression alleles (*Segregation Distortion* in *Drosophila melanogaster* [53], *Sex-Ratio* in *Drosophila simulans* [54]), to apparent drive systems with complete suppression [55]. This diversity in levels of drive suppression are seemingly unrelated to the age of the driver. A deeper understanding of the underlying factors explaining these marked differences would certainly help us design more effective synthetic gene drives.

In conclusion, we find that the *t-Sry* system is likely to be a relatively safe gene drive system, because it is non-transgenic, escapes will not lead to uncontrollable spread of drive, and interventions can be reversed simply by halting *t-Sry* releases. We find that rates of polyandry are highly likely to impact on the success of sperm-killing gene drive systems, and that understanding the rate of polyandry in target populations is important. However, we also argue that ancient, highly successful gene drive systems like the *t* haplotype may avoid the major risk to many synthetic drive systems, that of rapid evolution of complete suppression of the driver. We suggest that finding and understanding the mechanisms of ancient natural gene drives could lead to improvements in synthetic gene drive design.

Data accessibility. R and Maple source code of the model are available at the Dryad Digital Repository: https://doi.org/10.5061/dryad.4g1t4ht [56].

Authors' contributions. A.M., T.A.R.P. and A.S. conceived of the idea. S.C. and T.A.R.P. contributed to the development of the model. A.M. implemented the model and performed the computations. S.C. verified and helped to develop the analytical results. S.C., T.A.R.P. and A.S. helped with the interpretation of the model. All authors discussed the results and contributed to the manuscript.

Competing interests. We declare we have no competing interests.

Funding. A.M. and A.S. were funded by Swiss National Science Foundation Postdoc Mobility fellowships (A.M.: P300PA_177830 and P2ZHP3_161970, A.S.: P300PA_177906). T.A.R.P. was funded by the UK Natural Environment Research Council (NE/P002692/1).

Acknowledgements. We thank Anna Lindholm for her invaluable feedback on the manuscript. We thank Yevhen Suprunenko for discussions on analytical solutions. We are grateful to the editors and reviewers for their constructive feedback.

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
