## [Reviewer comments · Proceedings of the Royal Society B: Biological Sciences]

Review History

RSPB-2019-0852.R0 (Original submission)

Review form: Reviewer 1 (Luke Holman)

Recommendation

Accept with minor revision (please list in comments)

Scientific importance: Is the manuscript an original and important contribution to its field?

Good

General interest: Is the paper of sufficient general interest?

Good

Quality of the paper: Is the overall quality of the paper suitable?

Excellent

Is the length of the paper justified?

Yes

Should the paper be seen by a specialist statistical reviewer?

No

Do you have any concerns about statistical analyses in this paper? If so, please specify them explicitly in your report.

No

It is a condition of publication that authors make their supporting data, code and materials available - either as supplementary material or hosted in an external repository. Please rate, if applicable, the supporting data on the following criteria.

Is it accessible?

N/A

Is it clear?

N/A

Is it adequate?

N/A

Do you have any ethical concerns with this paper?

No

Comments to the Author

I wrote my review as comment on the MS, as unfortunately I am rather rushed for time. Also, I think the MS is almost ready for publication, so my comments are mostly brief. The only things that will take more than few minutes to fix are my comments about the implicit assumptions in the model, which I think you should address by acknowledging, discussing, and possibly modelling what happens when they are relaxed. For example, you assume that t-carriers have equal survival rates to wild-type mice, and that t males have equal mating success to wild types. I can imagine that t actually lowers survival and mating success, and that its effects on survival might be sex-specific (either due to chance in which t-linked genes mutated, or because t is transmitted through males more often than females and so it probably experiences stronger selection for male fitness). Please feel free to contact me if you have any questions about my comments - I hope they prove useful, and I apologise for doing the inevitable thing of asking for models! That seems to happen for almost every theory paper I write, and I know it's not always necessary or easy to do.

Best wishes,

Luke Holman

Review form: Reviewer 2

Recommendation

Major revision is needed (please make suggestions in comments)

Scientific importance: Is the manuscript an original and important contribution to its field?

Good

General interest: Is the paper of sufficient general interest?

Good

Quality of the paper: Is the overall quality of the paper suitable?

Good

Is the length of the paper justified?

Yes

Should the paper be seen by a specialist statistical reviewer?

Yes

Do you have any concerns about statistical analyses in this paper? If so, please specify them explicitly in your report.

No

It is a condition of publication that authors make their supporting data, code and materials available - either as supplementary material or hosted in an external repository. Please rate, if applicable, the supporting data on the following criteria.

Is it accessible?

N/A

Is it clear?

N/A

Is it adequate?

N/A

Do you have any ethical concerns with this paper?

No

Comments to the Author

In this paper, Manser and colleagues use mathematical models to explore the effect polyandry might have on the utility of rodent control using a synthetic gene drive that links the male-determining gene, *sry*, to the sperm-killing meiotic distorter, the *t*-haplotype.

Male, XY individuals carrying the *t-sry* gene construct are predicted to be fertile (I am unsure if this is known for sure - if it is then it would be good to include actual details on the fitness of those mice, see below) and will pass this gene construct onto ~90% of offspring. XX, individuals carrying *t-sry* are predicted to be male and sterile. Thus, in theory, at least, the release of *t-sry* individuals into a population could lead to some level of population suppression and maybe eventual eradication.

There are however numerous issues that may thwart the success of such a control scheme, with polyandry, where females mate with multiple males likely to be a significant limiter successful pest control. Prior work has shown that *t* haplotype individuals achieve low fertilisation outcomes when competing against wild type males. This impairment may limit their ability to pass on their gene drive to the next generation.

In general, I found the paper to be well researched and written. The key findings are: 1) that *t-sry* will not spread as a single release, even under a model where there is no fitness difference between *t-sry* and wildtype, and; 2) that polyandry and sperm competition will thwart any

control campaign focused around t-sry. These findings are probably predictable because earlier work effectively reached the same conclusion (Backus and Gross, 2016), but the current work is a useful extension to the ongoing discussions on the use of such tools and the natural barriers that may affect their utility.

The authors, take this negative and turn it into a positive, concluding that this lack of self-sustaining spread may still result in a useful and controllable tool because to be successful it will require regular releases. This view might be accurate, but I suspect it is a little bit optimistic on a cost-benefit front. Their models suggest you would need to release 3-6 times more individuals to achieve some form of control. The cost of doing that level of release is likely significant and the ecological damage of releasing such a large number of rodents substantial. I suspect the authors may wish to think on both these issues and perhaps rephrase some of their discussion, particularly around lines 306?

That said, my general view is positive. The work is good, albeit quite narrowly focused. Thus, I wondered if there might be value in extending the models to explore a few more parameters. Among the things I thought would be valuable to include are:

- 1) An explicit test of the effect of fitness differentials between t-sry and wildtype would be helpful. Presumably, if the fitness of t-sry individuals is reduced compared to wildtype, the success of sustained control drops even further. Some sensitivity analyses to explore this might be valuable? While prior work on the fitness of t-haplotype has been inconclusive (line 274), a range of fitness differentials could be investigated. If there are data on the fitness of any individuals carrying the synthetic t-sry construct that would also be valuable to include.
- 2) Consideration of a couple of resistance components and their effects on the modelling outcomes. Two forms of resistance seem pertinent. The first, given this, is a paper on polyandry, would be to explore aspects of behavioural resistance. Sperm competitiveness is factored in, but if strong pre-mating mate preference arises among females for wildtype over t-haplotype males, this would further impair the reproductive fitness of t-haplotype males. Presumably, there are some mate choice data from experiments on t-haplotype that could be used to explore this issue?
- 3) Is there value in considering other gene drive systems in comparison to the t-sry system? Obviously, we know more about the biology of this system, but perhaps some insights might emerge around the effects of polyandry on suppression control using gene drives in other systems?

A few specific points:

Line 247-288: I think you need a reference to the prior theoretical study here.

Line 261: I suspect you mean to cite Esvelt and Gemmell, 2017 here? Gemmell and Tompkins, 2017, deals with t-sry issues.

Line 306: I would encourage some thought on the economic costs of needing to release 3-6 times more animals to achieve control, and the likely ecological impact that might have, particularly in a conservation context.

Line 315: Resistance, particularly by non-homologous end joining, is a major barrier to CRISPR based gene drives, although there are molecular biological solutions to this problem. Natural gene drives may break less frequently, as the authors suggest. However, despite being present for millions of years, they are not present in all populations, which implies that there must be natural suppressors to thwart their spread. I think some brief mention of this type of population-based

resistance is likely warranted. In addition, the resistance can occur when the cargo genes mutate. While such events are rare, when you are dealing with pest populations numbering in the millions, they will happen. XX, t-sry* individuals, where sry is non-function may become fertile females, which could result in a rapid loss of population suppression.

Decision letter (RSPB-2019-0852.R0)

10-May-2019

Dear Dr Manser,

Your manuscript has now been peer reviewed and the reviews have been assessed by an Associate Editor. The reviewers' comments (not including confidential comments to the Editor) and the comments from the Associate Editor are included at the end of this email for your reference. As you will see, we are all agreed that this is a very interesting paper that should make a valuable contribution to the special issue on 'Natural and synthetic gene drives'. However the reviewers have made some excellent comments on the current version of the manuscript and we would like to invite you to revise your manuscript to address them.

Research ethics:

Use of animals and field studies:

If your study uses animals please include details in the methods section of any approval and licences given to carry out the study and include full details of how animal welfare standards

were ensured. Field studies should be conducted in accordance with local legislation; please include details of the appropriate permission and licences that you obtained to carry out the field work.

If you wish to submit your data to Dryad (<http://datadryad.org/>) and have not already done so you can submit your data via this link [http://datadryad.org/submit?journalID=RSPB&manu=\(Document not available\)](http://datadryad.org/submit?journalID=RSPB&manu=(Document%20not%20available)), which will take you to your unique entry in the Dryad repository.

Please submit a copy of your revised paper within three weeks. If we do not hear from you within this time your manuscript will be rejected. If you are unable to meet this deadline please let us know as soon as possible, as we may be able to grant a short extension.

With best wishes,
Loeske Kruuk
Editor
Proceedings B
<mailto:proceedingsb@royalsociety.org>

Associate Editor

Comments to Author:

This is an excellent MS that fits squarely within the Proceedings special issue. However, both referees felt that the scope of the findings would be greatly improved by adopting a broader modelling component to explore a few more parameters. While I realise that this requires additional work, I think it will be well worth the effort.

Reviewer(s)' Comments to Author:

Referee: 1

Comments to the Author(s)

I wrote my review as comment on the MS, as unfortunately I am rather rushed for time. Also, I think the MS is almost ready for publication, so my comments are mostly brief. The only things that will take more than few minutes to fix are my comments about the implicit assumptions in the model, which I think you should address by acknowledging, discussing, and possibly modelling what happens when they are relaxed. For example, you assume that t-carriers have equal survival rates to wild-type mice, and that t males have equal mating success to wild types. I can imagine that t actually lowers survival and mating success, and that its effects on survival might be sex-specific (either due to chance in which t-linked genes mutated, or because t is transmitted through males more often than females and so it probably experiences stronger selection for male fitness). Please feel free to contact me if you have any questions about my comments - I hope they prove useful, and I apologise for doing the inevitable thing of asking for models! That seems to happen for almost every theory paper I write, and I know it's not always necessary or easy to do.

Best wishes,

Luke Holman

Referee: 2

Comments to the Author(s)

In this paper, Manser and colleagues use mathematical models to explore the effect polyandry might have on the utility of rodent control using a synthetic gene drive that links the male-determining gene, *sry*, to the sperm-killing meiotic distorter, the t-haplotype.

Male, XY individuals carrying the t-*sry* gene construct are predicted to be fertile (I am unsure if this is known for sure - if it is then it would be good to include actual details on the fitness of those mice, see below) and will pass this gene construct onto ~90% of offspring. XX, individuals carrying t-*sry* are predicted to be male and sterile. Thus, in theory, at least, the release of t-*sry* individuals into a population could lead to some level of population suppression and maybe eventual eradication.

There are however numerous issues that may thwart the success of such a control scheme, with polyandry, where females mate with multiple males likely to be a significant limiter successful pest control. Prior work has shown that t haplotype individuals achieve low fertilisation outcomes when competing against wild type males. This impairment may limit their ability to pass on their gene drive to the next generation.

In general, I found the paper to be well researched and written. The key findings are: 1) that t-*sry*

will not spread as a single release, even under a model where there is no fitness difference between t-sry and wildtype, and; 2) that polyandry and sperm competition will thwart any control campaign focused around t-sry. These findings are probably predictable because earlier work effectively reached the same conclusion (Backus and Gross, 2016), but the current work is a useful extension to the ongoing discussions on the use of such tools and the natural barriers that may affect their utility.

The authors, take this negative and turn it into a positive, concluding that this lack of self-sustaining spread may still result in a useful and controllable tool because to be successful it will require regular releases. This view might be accurate, but I suspect it is a little bit optimistic on a cost-benefit front. Their models suggest you would need to release 3-6 times more individuals to achieve some form of control. The cost of doing that level of release is likely significant and the ecological damage of releasing such a large number of rodents substantial. I suspect the authors may wish to think on both these issues and perhaps rephrase some of their discussion, particularly around lines 306?

That said, my general view is positive. The work is good, albeit quite narrowly focused. Thus, I wondered if there might be value in extending the models to explore a few more parameters. Among the things I thought would be valuable to include are:

- 1) An explicit test of the effect of fitness differentials between t-sry and wildtype would be helpful. Presumably, if the fitness of t-sry individuals is reduced compared to wildtype, the success of sustained control drops even further. Some sensitivity analyses to explore this might be valuable? While prior work on the fitness of t-haplotype has been inconclusive (line 274), a range of fitness differentials could be investigated. If there are data on the fitness of any individuals carrying the synthetic t-sry construct that would also be valuable to include.
- 2) Consideration of a couple of resistance components and their effects on the modelling outcomes. Two forms of resistance seem pertinent. The first, given this, is a paper on polyandry, would be to explore aspects of behavioural resistance. Sperm competitiveness is factored in, but if strong pre-mating mate preference arises among females for wildtype over t-haplotype males, this would further impair the reproductive fitness of t-haplotype males. Presumably, there are some mate choice data from experiments on t-haplotype that could be used to explore this issue?
- 3) Is there value in considering other gene drive systems in comparison to the t-sry system? Obviously, we know more about the biology of this system, but perhaps some insights might emerge around the effects of polyandry on suppression control using gene drives in other systems?

A few specific points:

Line 247-288: I think you need a reference to the prior theoretical study here.

Line 261: I suspect you mean to cite Esvelt and Gemmell, 2017 here? Gemmell and Tompkins, 2017, deals with t-sry issues.

Line 306: I would encourage some thought on the economic costs of needing to release 3-6 times more animals to achieve control, and the likely ecological impact that might have, particularly in a conservation context.

Line 315: Resistance, particularly by non-homologous end joining, is a major barrier to CRISPR based gene drives, although there are molecular biological solutions to this problem. Natural

gene drives may break less frequently, as the authors suggest. However, despite being present for millions of years, they are not present in all populations, which implies that there must be natural suppressors to thwart their spread. I think some brief mention of this type of population-based resistance is likely warranted. In addition, the resistance can occur when the cargo genes mutate. While such events are rare, when you are dealing with pest populations numbering in the millions, they will happen. XX, t-sry* individuals, where sry is non-function may become fertile females, which could result in a rapid loss of population suppression.

Author's Response to Decision Letter for (RSPB-2019-0852.R0)

See Appendix A.

Decision letter (RSPB-2019-0852.R1)

26-Jul-2019

Dear Dr Manser

My apologies in the delay in sending you this decision due to summer travel.

I am pleased to inform you that your manuscript entitled "Controlling Invasive Rodents via Synthetic Gene Drive and the Role of Polyandry" has been accepted for publication in Proceedings B.

Open Access

Paper charges

Thank you for your very interesting contribution to the journal and the special issue. On behalf of the Editors of the Proceedings B, we look forward to your continued submissions to the Journal.

Yours sincerely,

Professor Loeske Kruuk
Editor, Proceedings B
mailto:proceedingsb@royalsociety.org

Associate Editor:

Comments to Author:

I am satisfied that the authors have addressed all the points raised by the referees. It is now an improved paper that should be of great interest to people working on gene drive systems.

Appendix A

Response to Reviewers' Comments

Associate Editor

This is an excellent MS that fits squarely within the Proceedings special issue. However, both referees felt that the scope of the findings would be greatly improved by adopting a broader modelling component to explore a few more parameters. While I realise that this requires additional work, I think it will be well worth the effort.

Response 1: Many thanks, we agree about the broader modelling, and have now added an analysis on both survival differences and mate choice (see Response 2 for more details).

Referee: 1

I wrote my review as comment on the MS, as unfortunately I am rather rushed for time. Also, I think the MS is almost ready for publication, so my comments are mostly brief. The only things that will take more than few minutes to fix are my comments about the implicit assumptions in the model, which I think you should address by acknowledging, discussing, and possibility modelling what happens when they are relaxed. For example, you assume that t-carriers have equal survival rates to wild-type mice, and that t males have equal mating success to wild types. I can imagine that t actually lowers survival and mating success, and that its effects on survival might be sex-specific (either due to chance in which t-linked genes mutated, or because t is transmitted through males more often than females and so it probably experiences stronger selection for male fitness). Please feel free to contact me if you have any questions about my comments - I hope they prove useful, and I apologise for doing the inevitable thing of asking for models! That seems to happen for almost every theory paper I write, and I know it's not always necessary or easy to do.

Best wishes,

Luke Holman

Response 2. Thanks for the constructive and helpful feedback. We have comprehensively addressed all your pdf comments (see section just below).

As for the broader points about survival differences and mate choice (which were echoed by both the associate editor and referee 2)- we agree that both effects will central to predict the success of the construct. We have thus:

a) added an analysis of genotype-specific survival and mate choice to the supplementary material [Text S4, Figure S3, Figure S4] and

b) extended and reworded the introduction to hopefully highlight these two effects more clearly [LL92-93, LL149-150, LL168-169, LL293-294]

The reason why we have not added the analyses to the main manuscript, and why we have not examined them previously are threefold. First, the empirical evidence for both survival differences and mate choice are not as clear, strong, and robust as the polyandry effect. Surprisingly, despite the size and lack of recombination of the t haplotype, there is the evidence for (heterozygote) survival cost of the t haplotype is highly inconsistent, with several studies even suggesting the t increases survival (e.g. Manser et al 2011 Evolution, Auclair et al 2013 PlosOne and references within). The evidence for avoiding mating with t males is stronger, but still only finds relatively weak preferences, and in some studies, no avoidance at all (reviewed in Manser, König & Lindholm 2014 JEB). Second, survival differences have been examined comprehensively by a previous modelling study on the system (Backus and Gross 2016 Ecosphere). Third, rather than developing a model that aims to create realistic scenarios of t-Sry release campaigns, the main goal our current MS was to create a conceptual model that highlight/examines an effect in polyandry that we a) deem quantitatively the strongest in the t haplotype system based on empirical evidence and b) has so far been neglected with respect to synthetic drive applications. That said, we do appreciate that both effects are potentially important and broaden the scope of our model. Hence the addition to the supplementary, but not the main text.

PDF comments

L32. raises

Response 3. Changed [L31]

Overall. make refs chronological and use Proc B style

Response 4. All references are now chronological. We kept the reference style for the time being as we find it more convenient to see the authors/year directly during the review

process. This would surely be dealt with during production, as we will submit the MS as a Latex/Bibtex file.

L75. *some* of the costs. They will still suffer from the linked mutations

Response 5. Changed [L74]

LL85-86. What processes? Don't mention here if you want to return to this later.

Response 6. This is sentence was meant to introduce / set the stage for the subsequent paragraph. We have now moved the sentence to the next paragraph, where it belongs, and changed "processes" to "mechanisms" (as used in the following sentence) to make the link more explicit [LL85-86]

You haven't mentioned yet that t is a massive region (like 5-10% of the genome isn't it?). So, the main cost is likely to be A) the 50% dead sperm, and B) the many deleterious mutations inside t. The actual Sry insert is likely to be relatively unimportant for male fitness (obvs it is bad for females). None of this information is very clear from this passage. The non-t genetic background will be rapidly lost assuming the first release of mice manages to breed, and you could always backcross the lab-bred t-Sry mice to wild mice before releasing them

Response 7. Agreed. Interestingly, as mentioned above, the main costs of the t (given its potential to accumulate deleterious mutations) seem to occur in homozygotes (lethality/sterility). Heterozygote fitness effects, on the other hand, have been measured across various contexts / populations, but show a far less obvious picture. In some cases, the t has actually been found to give mice a fitness advantage. We have reworded this section, hopefully clarifying it [L92-93]. By the way, the fact the t's main fitness effect appears to be recessive (mainly present in t/t homozygotes) could be the result of an observation bias. Drivers with large costs to both heterozygotes and homozygotes will probably disappear from populations rather quickly. Equivalently, drivers with few homozygote costs, on the other hand, might readily spread to fixation. We may thus be bound to observe old drivers we strong homozygote, but potentially weaker heterozygote costs.

LL103-104. I suggest: causes the t haplotype to be transmitted to more than 90% of the offspring

Response 8. Changed [L106-107]

L105. ...the sperm of +/t males makes these males...

Response 9. *Changed [L108]*

Intro. You have not yet mentioned if Sry manages to reach high frequencies in nature (I think it's like 5-10%?). This suggests that indeed it is prevented from spreading by its various costs - and that's BEFORE you make it dominant sterile/lethal to females... I suspect this is not going to work, but I look forward to the Results :)

Response 10. *Do you mean t haplotype frequencies? We have now added a sentence the mentions the low, stable naturally observed frequencies, and polyandry currently being our best explanation for them [L116-117]*

L126. Maybe instead use + as in the intro. This also avoids confusion bc w usually means fitness

Response 11. *This has been changed, (hopefully) throughout the manuscript.*

LL131-132. Why does it matter that ww cannot have tt offspring? You still need to track them

Response 12. *We apologise for the confusion. The only reproductive females are XX++. So all offspring will carry at least one X and one + allele. Thus, tt individuals cannot occur in the population at any point in the life cycle. Adding lines calculating these frequencies to the model would be uninformative, as they be zero at all life stages. So the inclusion of tt offspring in the model will have no impact on the outcome of the model. While we understand it might be nice to include them for the sake of completeness, we think doing this would unnecessarily complicate the presentation of the model, and make it harder for readers to understand. We now change the line to "As all reproductive females are XX++, tt homozygotes cannot feature at any life stage (Figure 1). This leaves us with four genotypes to track, (...)" [L135-137]*

LL136-138. Why the f(x) notation? These are not functions, they are counts

Response 13. *The f() here were meant to denote 'frequency', but agreed, they are confusing, and have been removed [L141-143]*

Methods. The equal mortality of W and D individuals assumption seems a stretch given that t carries lots of bad mutations. Would it be hard to have a separate mortality rate for W and D?

Do you think it would interact with polyandry (I know the basic conclusion is obvious - increasing mortality of D individuals would be bad - but it might also affect your polyandry conclusion)

Response 14. *We have now added the analysis of differential survival rates between genotypes (Text S4, also see Response 2). Therein, we also address the question of how survival and mate choice interact with polyandry. We found very little evidence for interactive effects, although this conclusion may depend on the underlying modelling assumptions / implementation details [Figure S3 and S4].*

Methods. All your models assume (implicitly) that + and t males are equally likely to mate, right? Is that justified?

I know females cannot discriminate between these males in a 2-choice trial, but sexual selection has more dimensions than that - e.g. t males might spend less time mate-searching if they are less healthy.

Please either state this assumption, relax it, and/or discuss its plausibility and evolutionary consequences. I feel like it will affect your “3- to 6-fold” result a bit, and will interact with psi (polyandry parameter).

You also implicitly assume that the cost of Sry-t is the same in males in females (i.e. it zero). What if it has a sex-specific effect on mortality? Would that matter? Again, you could either model or discuss this.

Response 15. *We have now added the analysis of both mate choice and genotype-specific survival effects which incorporates the survival cost/benefit of t-Sry, being a male, and carrying an extra copy of the Sry gene (see Response 2).*

L180. Instead I suggest “...depends on the frequency of t-Sry”

Response 16. *Changed [L187]*

L193. delete “the”

Response 17. *Deleted*

L200. replace with “...to maintain the same frequency as in the previous generation”

Response 18. *Changed [L207]*

L248. cite it

Response 19. *Reference has been added [L266].*

L250. Pose

Response 20. *Changed [L268]*

LL310-311. Also, didn't they find that it didn't really work in male mice, because only females seem to use the homology- directed repair pathway? Males prefer NHEJ

Response 21. *That's correct, we now say "in female mice" [L347].*

Discussion. Also, it doesn't require any non-mouse genes to be added, so it is arguably not a transgenic animal, which makes it an easier sell

Response 22. *Excellent point. We have added a sentence and a sentence fragment in the final paragraph where we mention this [L337-339, L377].*

Referee: 2

In this paper, Manser and colleagues use mathematical models to explore the effect polyandry might have on the utility of rodent control using a synthetic gene drive that links the male-determining gene, *sry*, to the sperm-killing meiotic distorter, the t-haplotype.

Male, XY individuals carrying the t-*sry* gene construct are predicted to be fertile (I am unsure if this is known for sure - if it is then it would be good to include actual details on the fitness of those mice, see below) and will pass this gene construct onto ~90% of offspring. XX, individuals carrying t-*sry* are predicted to be male and sterile. Thus, in theory, at least, the release of t-*sry* individuals into a population could lead to some level of population suppression and maybe eventual eradication.

There are however numerous issues that may thwart the success of such a control scheme, with polyandry, where females mate with multiple males likely to be a significant limiter successful pest control. Prior work has shown that t haplotype individuals achieve low fertilisation outcomes when competing against wild type males. This impairment may limit their ability to pass on their gene drive to the next generation.

In general, I found the paper to be well researched and written. The key findings are: 1) that t-sry will not spread as a single release, even under a model where there is no fitness difference between t-sry and wildtype, and; 2) that polyandry and sperm competition will thwart any control campaign focused around t-sry. These findings are probably predictable because earlier work effectively reached the same conclusion (Backus and Gross, 2016), but the current work is a useful extension to the ongoing discussions on the use of such tools and the natural barriers that may affect their utility.

The authors, take this negative and turn it into a positive, concluding that this lack of self-sustaining spread may still result in a useful and controllable tool because to be successful it will require regular releases. This view might be accurate, but I suspect it is a little bit optimistic on a cost-benefit front. Their models suggest you would need to release 3-6 times more individuals to achieve some form of control. The cost of doing that level of release is likely significant and the ecological damage of releasing such a large number of rodents substantial. I suspect the authors may wish to think on both these issues and perhaps rephrase some of their discussion, particularly around lines 306?

Response 23: *Thanks for the helpful and detailed feedback. This is a valuable suggestion. We now try to give the reader a better impression of what a 3- to 6-fold increase could mean ecologically. We now dedicate an entire result section [LL252-262] and a new figure (Figure 3) to the examination of the ecological and economical impacts of release campaigns. It shows a) the maximum population size attained during the release, b) the total number of animals released, and c) the time to population eradication. We find that polyandry has little impact on the maximum population size reached during the intervention (no more than 1% more than monandry). In contrast, the required release rate to cause extinction, as well as the overall number of animals released, increases significantly with higher polyandry. However, we note that even with high polyandry, release rates of ~20% of the population size are sufficient to swamp any polyandry, choice and survival effects, and appear to always result in population extinction. Note that we deem these new results more informative and relevant to the main message of the paper than (previous) Figure 2. We thus moved (previous) Fig. 2 into the Supplementary (now Figure S5), and replaced it with Figure 3.*

In terms of economic costs, our model cannot realistically provide a good estimate. In an actual intervention, the economic costs are going to be highly dependent on the details of the system. For example, how long females live, how many offspring they have, and the population density (a proxy for sperm competitive environment) on that particular target island will all matter. The details of the intervention will also matter. For an isolated island

that is difficult to reach, a large input of tsry males once each year might be the only option. Alternatively, an inhabited island could potentially have a breeding facility set up on the island, allowing regular smaller releases. Current island exterminations vary enormously in their costs, often due to factors like isolation, the regulatory framework, and mitigation costs (see Holmes et al 2015 Biological Invasions for a discussion). We have now added a paragraph discussing these issues into the Discussion [L334-344]

That said, my general view is positive. The work is good, albeit quite narrowly focused. Thus, I wondered if there might be value in extending the models to explore a few more parameters. Among the things I thought would be valuable to include are:

1) An explicit test of the effect of fitness differentials between t-sry and wildtype would be helpful. Presumably, if the fitness of t-sry individuals is reduced compared to wildtype, the success of sustained control drops even further. Some sensitivity analyses to explore this might be valuable? While prior work on the fitness of t-haplotype has been inconclusive (line 274), a range of fitness differentials could be investigated. If there are data on the fitness of any individuals carrying the synthetic t-sry construct that would also be valuable to include.

Response 24. *We have now added the analysis on the impact of genotype-specific survival rates on the release dynamics (Text S4, see Response 2). However, the empirical data on survival differences is not as clear as one might perhaps expect. Several studies have in fact found a +/t heterozygote survival advantage over +/+ wildtypes (e.g. Manser et al 2011 Evolution, Auclair et al 2013 PlosOne and references within). It is likely that both the exact t strain used, and the genetic background on which it is found, will interact to determine the fitness costs and benefits of the t. The current evidence also suggests that the costs and benefits are likely to be sex specific. Ultimately, any potential use will need robust measures on survival differences of the t-Sry construct on the target population itself.*

2) Consideration of a couple of resistance components and their effects on the modelling outcomes. Two forms of resistance seem pertinent. The first, given this, is a paper on polyandry, would be to explore aspects of behavioural resistance. Sperm competitiveness is factored in, but if strong pre-mating mate preference arises among females for wildtype over t-haplotype males, this would further impair the reproductive fitness of t-haplotype males. Presumably, there are some mate choice data from experiments on t-haplotype that could be used to explore this issue?

Response 25. *We have now added an analysis where we examine the effect of mate choice against +/t heterozygotes, as well as potential interaction effects with polyandry (Text S4,*

see Response 2). Mate choice has indeed been measured repeatedly in this system. Some studies, particularly on one population in the late 80ies, have reported olfactory avoidance of +/t males, although the evidence for avoidance in this population was only seen in t females, with wild-type females showing inconsistent preferences (reviewed in Lenington et al. 1992). However, other studies do not find an effect (Manser et al. 2015, Sutter & Lindholm 2016). We mention this in the discussion [LL294-296].

3) Is there value in considering other gene drive systems in comparison to the t-sry system? Obviously, we know more about the biology of this system, but perhaps some insights might emerge around the effects of polyandry on suppression control using gene drives in other systems?

Response 26: *We expect the effect of polyandry to be particularly relevant for synthetic constructs that use sperm killers (as this sperm damage creates the polyandry effect). However, it potentially also relevant in CI inducing endosymbiont drive (see Hoffmann's work), and some chromosome shredders may also reduce sperm competitive ability of their carriers. Thus, in theory, any gene drive that reduces sperm competitive ability could be affected, even if this reduction is a byproduct of general fitness costs of drive, rather than the drive mechanism itself. We mention this now in the discussion [L307-310].*

Moreover, we have attempted to broaden the discussion to other synthetic drivers by discussing their potential vulnerabilities to various forms of drive resistance. As argued on LL315-39, we speculate that sperm killers maybe less vulnerable to molecular resistance than homing based drivers (which are currently the most widely discussed), or, at least, that the behavioural forms of drive resistance (polyandry, mate choice) do not pose an insurmountable hurdle to release campaigns.

A few specific points:

Line 247-288: I think you need a reference to the prior theoretical study here.

Response 27: *Reference has been added [L266].*

Line 261: I suspect you mean to cite Esvelt and Gemmell, 2017 here? Gemmell and Tompkins, 2017, deals with t-sry issues.

Response 28: *Good point, the reference has been changed [L280].*

Line 306: I would encourage some thought on the economic costs of needing to release 3-6 times more animals to achieve control, and the likely ecological impact that might have, particularly in a conservation context.

Response 29: *We have now added a paragraph where we make some ecological and economical considerations, on the system overall, and with respect to the 3-6-fold polyandry-related increase [LL252-262, LL334-344, new Figure 3].*

Line 315: Resistance, particularly by non-homologous end joining, is a major barrier to CRISPR based gene drives, although there are molecular biological solutions to this problem. Natural gene drives may break less frequently, as the authors suggest. However, despite being present for millions of years, they are not present in all populations, which implies that there must be natural suppressors to thwart their spread. I think some brief mention of this type of population-based resistance is likely warranted. In addition, the resistance can occur when the cargo genes mutate. While such events are rare, when you are dealing with pest populations numbering in the millions, they will happen. XX, t-sry* individuals, where sry is non-function may become fertile females, which could result in a rapid loss of population suppression.

Response 30: *Point taken. The factors that explain the natural variation in drive frequency across populations is still poorly understood, and many drive suppression mechanisms may yet to be discovered. At least in the house mouse case, we think that polyandry may play a central role in drive suppression. In the absence of active release, polyandry does in fact predict drive extinction for realistic polyandry rates. Also, t frequencies are typically lower in populations with higher density, which would be expected if polyandry rates are higher in denser populations. So, in a sense, your argument was precisely the motivation for our entire study. Moreover, the fact that there is a distribution of t frequencies across population (rather than presence or absence) seems to support our point that the factors that suppress drive in the mouse system, whichever they may be, are quantitative rather than qualitative in nature. As a result, they may be surmountable by increasing release effort. The mutation of cargo is a good point that we now mention it in the discussion [L358-359].*